# Erythroprotective Potential of Phycobiliproteins Extracted from *Porphyridium cruentum*

**DOI:** 10.3390/metabo13030366

**Published:** 2023-03-01

**Authors:** Rubria Lucía Peña-Medina, Diana Fimbres-Olivarría, Luis Fernando Enríquez-Ocaña, Luis Rafael Martínez-Córdova, Carmen Lizette Del-Toro-Sánchez, José Antonio López-Elías, Ricardo Iván González-Vega

**Affiliations:** 1Department of Scientific and Technological Research, University of Sonora, Blvd Luis Encinas y Reforma S/N, Col. Centro, Hermosillo 83000, Mexico; 2Department of Research and Postgraduate in Food, University of Sonora, Blvd Luis Encinas y Rosales S/N, Col. Centro, Hermosillo 83000, Mexico; 3Department of Medical and Life Sciences, Cienega University Center (CUCIÉNEGA), University of Guadalajara, Av. Universidad 1115, Lindavista, Ocotlán 47820, Mexico

**Keywords:** antihemolytic, erythroprotective effect, phycobiliproteins, red blood cells, pigment-rich extracts, microalgae

## Abstract

There are multiple associations between the different blood groups (ABO and RhD) and the incidence of oxidative stress-related diseases, such as certain carcinomas and COVID-19. Bioactive compounds represent an alternative to its prevention and treatment. Phycobiliproteins (PBP) are bioactive compounds present in the microalga *Porphyridium cruentum* and, despite its antioxidant activity, their inhibitory effect on hemolysis has not been reported. The aim of this work was to evaluate the erythroprotective potential of phycobiliproteins from *P. cruentum* in different blood groups. The microalga was cultured in F/2 medium under controlled laboratory conditions. Day 10 of culture was determined as the harvest point. The microalgal biomass was lyophilized and a methanolic (MetOH), Tris HCl (T-HCl), and a physiological solution (PS) ultrasound-assisted extraction were performed. Extract pigments were quantified by spectrophotometry. The antioxidant activity of the extracts was evaluated with the ABTS^+•^, DPPH^•^, and FRAP methods, finding that the main antioxidant mechanism on the aqueous extracts was HAT (hydrogen atom transfer), while for MetOH it was SET (single electron transfer). The results of the AAPH, hypotonicity, and heat-induced hemolysis revealed a probable relationship between the different antigens (ABO and RhD) with the antihemolytic effect, highlighting the importance of bio-directed drugs.

## 1. Introduction

There are multiple associations between the different blood groups of the ABO and RhD system with the incidence of diseases, like certain carcinomas and COVID-19 (Table 1 and Table 2). Its relevance lies in the presence of ABO antigens in the erythrocyte membrane, but also as circulating solubilized glycoproteins in secretions and excretions, epithelium, platelets, and endothelium [1]. The treatment and prevention of these diseases has been oriented towards the study of bioactive compounds capable of succumbing to the effects of oxidative stress [2], which is defined as an imbalance between antioxidant systems and reactive nitrogen, sulfur, carbonyl, and oxygen species [3], classified as radical and non-radical [4].

Free radicals, or just radicals, are chemical species that have at least one unpaired electron in their last energy orbital, which is the product of a homolytic break [5]. Free radicals can alter components of erythrocyte cell membranes, particularly phospholipids, which leads to hemolysis, mainly induced by peroxyl radicals [6]. Hemolysis involves the rupture of the erythrocyte cell membrane, releasing hemoglobin and other intracellular constituents into the plasma [7]. When hemolysis persists, it can result in hemolytic diseases and conditions related to oxidative stress [8].

**Table 1 metabolites-13-00366-t001:** Associations between the incidence of diseases with the blood groups of the ABO system.

ABO Group	Associated Disease	Reference
A	Breast cancer	[9]
	Pancreatic cancer	[10]
	COVID-19	[11]
	Cholelithiasis	[12]
	Thrombosis (pulmonary embolism and deep vein thrombosis)	[12]
	SARS	[1]
B	Pancreatic cancer	[10]
	Blood disorders (gastric ulcer and duodenal ulcer)	[12]
	Thyroid cancer	[12]
	Diabetes mellitus type 2	[13]
O	Thyrotoxicosis	[12]
	Gestational hypertension	[12]
	Renal and ureteral calculi	[12]
	Blood disorders (gastric and duodenal ulcer)	[12]
AB	Squamous cell carcinoma	[14]
	Cholelithiasis	[12]
	Thyroid cancer	[12]
	Colorectal cancer	[15]

**Table 2 metabolites-13-00366-t002:** Associations between the incidence of diseases with the blood groups of the RhD system.

RhD Group	Associated Disease	Reference
Positive	Breast cancer	[9]
	Gestational hypertension	[12]
	COVID-19	[16]
Negative	Small cell lung cancer	[17]

Oxidative stress triggers actions that affect important biomolecules, including DNA degradation, lipid peroxidation, and proteolysis. It is associated with non-communicable diseases, such as cardiovascular diseases and cancer [18]. Noncommunicable diseases were the cause of 74% of all deaths worldwide in 2019, meaning a 40.805 million deaths [19]. Therefore, health sector intervention, as well as associated oxidative stress research, are urgent.

Bioactive compounds represent an alternative to prevent and counteract the effects of oxidative, specifically, the antioxidant compounds [2]. Antioxidants are agents who inhibit or delay the oxidation of biologically relevant molecules in order to keep cells in a lower redox state [20]. Its mechanism of action involves its direct reactivity with free radicals, a phenomenon called antiradical activity, as well as its ability to eliminate free radicals, a quality designated as antioxidant activity [21]. There are two free radical inhibition strategies: SET and HAT. SET methods detect the capacity to transfer an electron to reduce a compound. SET reactions are used to exhibit a color change corresponding to the oxidant agent reduction [22]. Some methods based on SET are the Folin–Ciocalteu, FRAP, and CUPRAC assays [23]. The techniques based on HAT measure the ability of an antioxidant to neutralize free radicals by donating a hydrogen atom. In HAT reactions, the bond dissociation energy of the hydrogen donor group and the ionization energy of the candidate antioxidant are the main attributes influencing their relative reactivity. They usually involve the use of a fluorescent probe that can react with free radicals or with antioxidants. These techniques include ORAC, TRAP, TOSC, and Crocin bleaching tests. Mixed assays generally involve the sequestration of a chromatophore or fluorophore radical by antioxidants in which the HAT and SET mechanisms are involved to a variable extent, depending on the reaction conditions. The main mixed tests include the ABTS^+•^, DPPH^•^, and DMPD^•+^ [22,23].

Currently, there is a growing trend towards the search for natural antioxidants [24]. Oceans are a promising source of natural resources, due to their extension, minimal exploration, and biological richness [25], including microalgae, a polyphyletic group of photosynthetic aquatic microorganisms that have shown excellent performance in the production of bioactive compounds as phycobiliproteins, which stand out for their diverse applications and therapeutic activities [26].

Phycobiliproteins (PBP) are microalgal accessory photosynthetic pigments present in cyanobacteria (e.g., *Arthrospira* sp.), cryptophytes (e.g., *Cyanophora* sp.), glaucophytes (e.g., *Cyanophora* sp.), and rhodophytes (e.g., *Porphyridium* sp.). They configure complexes called phycobilisomes, inserted in the thylakoid membrane of the chloroplast. The main bioactivities recognized for PBP are anti-inflammatory, immunostimulant, anti-radical, and antioxidant [27,28,29]. *Porphyridium cruentum* is a marine planktonic red microalga that appears as an excellent source of PBP, categorized by their spectroscopic characteristics as B-Phycoerythrin (B-PE; λmax = 545–565 nm), R-Phycocyanin (R-PC; λmax = 635–638 nm), and allophycocyanin (APC; λmax = 650–660 nm) [28,30].

Between the PBP bioactivities, the antioxidant, antiviral, antitumoral, immunostimulant, and anti-inflammatory properties were present [28,31,32,33]. PBP antioxidant activity implies the free radical’s inhibition capacity because they can sequester reactive oxygen species (ROS) and reactive nitrogen species (NRS). In addition, they have the quality of delaying or inhibiting the oxidation of lipids (lipid peroxidation) and proteins (proteolysis) in the plasma membrane [28]. PC is a chelating agent for ferric ions [34], which are involved in many free radical generation processes. The hepatoprotective, anti-inflammatory, and anticancer effects are linked to its strong antioxidant capacity [35].

The demand for new bioactive compounds for the treatment of chronic-degenerative diseases, microbial infections, and inflammatory processes [36] justifies the importance of exploring the PBP biological activities not previously reported, such is the case of its potential erythroprotective effect on free radical-induced hemolysis. Due to the above, the aim of this study was to evaluate the erythroprotective potential of phycobiliproteins from *Porphyridium cruentum*.

## 2. Materials and Methods

### 2.1. Biological Material

The red microalga *Porphyridium cruentum* were obtained from the microalgae collection of the Departamento de Investigaciones Científicas y Tecnológicas (DICTUS) from the University of Sonora.

The red blood cells (RBC) with different antigens from the system ABO and RhD were collected from healthy adult volunteers of 20–50 years old with prior consent by venipuncture and were stored in tubes with spray-coated K_2_EDTA. The erythrocyte suspension was prepared at 10% with approximately 4.7 to 6.1 × 10^6^ cells·μL^−1^ and was processed for analysis immediately after extraction.

All processes using human blood were performed following the Mexican (NOM-253-SSA1-2012) and international (FDA: CFR—Code of Federal Regulations Title 21, part 640 Additional Standards for human blood and blood product, Support. B Red blood cells, Sec. 640.14 Testing the blood [21CFR640.14]) regulations. The laboratory where the extraction took place is accredited by ISO-IEC 17025 (NMX-EC-17025) and ISO 15189 elaborated by technical committee ISO/TC 212 (Clinical Laboratory Testing and In vitro Diagnostic Systems), taking ISO/IEC 17025 and ISO 9001 as reference.

The suspension was centrifuged at 1500 rpm for 10 min and washed three times with isotonic solution. RBC suspension was used in the biocompatibility blood assay and in the evaluation of erythroprotective potential from the extracts of *P. cruentum*.

### 2.2. Microalgal Culture and Experimental Design

*P. cruentum* were harvested in 10 L tubes with F/2 medium [37] at 35 UPS (Practical Salinity Unity). The culture was maintained under laboratory-controlled conditions at constant aeration and temperature (20 ± 1 °C). Illumination was provided by tubular fluorescent lamps in a continuous photoperiod (24 h), with constant light quality (white light; λ = 400–750 nm) and intensity (35 μmol·m^−2^·s^−1^). The microalga were harvested at the 10th day of culture, which represents the initial day of the logarithmic phase. The total biomass collected was recovered by centrifugation and lyophilized (Yamato Scientific C.O., L.T.D. Japan) for subsequent tests.

A completely randomized single experimental design was performed with a minimum of three replications for each analysis (n ≥ 3).

### 2.3. Growth Kinetics

At least two daily counts were performed in quadruplicate 250 mL flasks, for 13 days, to estimate the cell concentration and growth kinetics of *P. cruentum*. For this, daily samples of the microalgal culture were taken from the flasks, which were fixed with a Lugol’s solution (I_2_ 1% and KI 2% in distilled water).

The counts took place in a 0.1 mm Neubauer chamber, under a Carl Zeiss Axiostar plus model 1169–149 compound optical microscope. The data on the number of cells per quadrant were entered into an Excel database. The growth kinetic curve of the microalga was established from the daily cell concentration. Cell concentration (C; number of cells·mL^−1^) was obtained with the Equation (1) based on the average of cells (n, number of cells) and the conversion factor (0.004 mm^3^) [38]:C = n/(4 × 10^−6^)(1)

### 2.4. Phycobiliproteins Extraction

The MetOH extract was obtained by mixing 0.5 g of lyophilized microalgae with 25 mL of 99% methanol to remove chlorophylls and carotenoids. Ultrasound-assisted extraction was performed at 400 W ultrasonic power 3 pulses, 500 mHz, and 15 s per pulse with an amplitude of 30%. Following that, it was maintained in darkness for 24 h. The sample was centrifugated (Beckman model J2-21) for 15 min at 4 °C and 2000 rpm. The extraction was repeated. The same procedure was carried out with PS and T-HCl to obtain aqueous extracts rich in PBP.

### 2.5. Pigments and Protein Quantification

The pigments were partially quantified by spectrophotometry with the proposed equations from their respective molar extinction coefficient in a 96-well microplate spectrophotometer (Multiskan Go, Thermo Scientific, Waltham, MA, USA). The concentration of the different phycobiliproteins (APC, B-PE, and R-PC) in the aqueous extracts was estimated following the protocols [39,40,41] (Equations (2)–(4)). Additionally, the concentration of total chlorophylls (C_total_) and total carotenoids (C_x+c_) of the methanolic extract was estimated following the methodology published [42,43] (Equations (5) and (6)). The technique was standardized using MetOH, PS, and T-HCl as blanks. For the determinations, 300 µL of extract szx placed in a 96-well microplate.
R-PC = (A620 − 0.7 × A565)/7.38(2)
APC = (A560 − 0.19 × A620)/5.65(3)
B-PE = (A565 − 2.8 × [R-PC] − 1.34 × [APC])/12.7(4)
C_total_ = (21.3877 × A630) + (10.3739 × A647) + (10.3739 × A664) + (5.5309 × A691)(5)
C_x+c_ = ((1000.65 × A470) − (2.86 × Ca) − (129.2 × Cb))/221(6)

Protein concentration was estimated quantitatively by the bicinchoninic acid assay (BCA) [44]. It is a colorimetric technique that allows for the detection and quantification of total proteins, combining the reduction of Cu^+2^ to Cu^+1^ by proteins in an alkaline medium (Biuret reaction) with a high sensitivity and selectivity of colorimetric detection of cations of copper (Cu^+1^) using a single reagent containing bicinchoninic acid. The purple-colored reaction produced by this reaction is caused by the chelation of two copper-binding BCA molecules. Dilutions were made from a standard albumin solution (BSA) (2 mg/mL). A 1:100 dilution of the aqueous extracts was made. The reaction was carried out in a 96-well microplate, and 200 µL of the WR reagent was added to each well. The reaction was incubated at 25 °C for 16 h. Optic density was taken at 562 nm.

### 2.6. Analysis of the Biological Activities of Phycobiliproteins

Different bioassays were carried out to analyze the biological activities of the aqueous extracts rich in PBP from PS, T-HCl, and MetOH extracts, including the determination of antioxidant activity (ABTS^•+^, DPPH^•^, and FRAP), the biocompatibility assay, the erythroprotective potential (antihemolytic activity, hypotonicity, and heat-induced hemolysis inhibition).

### 2.7. Determination of Antioxidant Activity

The antioxidant activity of the extracts (PS, T-HCl, and MetOH) was determined by evaluating the antiradical potential (ABTS^•+^ and DPPH^•^ inhibition assay) and the reducing potential (FRAP). A Trolox curve was made to express the results of the different antioxidant activity assays as µmol TE (Trolox equivalents)/g DW.

#### 2.7.1. Inhibition Assay of 2,2′-azinobis-(3-ethylbenzothiazolin)-6-sulfonic acid Free Radical (ABTS^+•^)

Antiradical activity against the free radical ABTS^+•^ was analyzed based on the technique described with some modifications [45]. The ABTS salt (19.3 mg) was dissolved in distilled water (5 mL), then 88 μL of potassium persulfate solution (K_2_S_2_O_8_) (0.0378 g/L) was added and left to stand in the dark for 12 h to oxidize ABTS as ABTS^+•^. The ABTS^+•^ solution was diluted with 99% ethanol until an absorbance of 0.7 ± 0.05 at 734 nm was obtained. In a 96-well microplate, the following treatments were carried out by triplicate: negative control (20 µL of ethanol + 270 µL of ABTS^•+^), standard control (20 µL of standard antioxidant + 270 µL of ABTS^•+^) and samples (20 µL of extract + 270 µL of ABTS^•+^). It was allowed to stand for 30 min in the darkness at room temperature. Subsequently, the readings were taken at 734 nm in a microplate spectrophotometer (Multiskan Go, Thermo Scientific, Waltham, MA, USA).

#### 2.7.2. Free Radical Inhibition Assay 1,1-diphenyl-2-picrylhydrazyl (DPPH^•^)

The radical inhibitory capacity of the free radical DPPH^•^ was evaluated following the methodological protocol published [46]. The methanolic solution of DPPH^•^ (6 × 10^−5^ mol·L^−1^) was prepared and 99% methanol was gradually added until an adjusted absorbance of 0.7 ± 0.05 at 515 nm was obtained. The following treatments were carried out by triplicate in a 96-well microplate: negative control (20 µL of methanol + 200 µL of DPPH^•^), standard control (20 µL of standard antioxidant + 200 µL of DPPH^•^), and sample (20 µL of extract + 200 µL of DPPH^•^). It was left to stand for 30 min at room temperature in the dark. Absorbances at 515 nm were taken in a spectrophotometer (Multiskan Go, Thermo Scientific, Waltham, MA, USA).

#### 2.7.3. Ferric Ion Antioxidant Reducing Power (FRAP)

The reducing potential was determined following the methodology proposed, with some modifications [47]. Stock solutions were sodium acetate buffer (300 mM, pH 3.6), ferric chloride (FeCl3) solution (20 mM), and TPTZ (2,4,6-tripidyl-s-triazine) solution (10 mM) in HCl (40mM). The FRAP working solution was prepared in a ratio of 10:1:1 (buffer:FeCl3:TPTZ). The following treatments were carried out by triplicate in a 96-well microplate: negative control (20 µL of methanol + 280 µL of FRAP), standard control (20 µL of standard antioxidant + 280 µL of FRAP) and sample (20 µL of extract + 280 µL of FRAP). It was allowed to stand for 30 min in the dark at room temperature. Optical densities at 638 nm were obtained every 10 min for one hour in a spectrophotometer (Multiskan Go, Thermo Scientific, Waltham, MA, USA).

### 2.8. Biocompatibility Blood Assay

The biocompatibility blood assay consists of a direct hemolysis induction in order to evaluate the microalgal extracts’ cytotoxicity. The methodology described was followed [48]. Different treatments were incubated at 37 °C for 2, 4, and 6 h: negative control (150 µL erythrocytes + 150 µL PBS), positive control (150 µL erythrocytes + 150 µL Triton-X 100 at 1%), and sample (150 µL erythrocytes + 150 µL extract). Following that, 1 mL of PBS was added and centrifuged at 1500 rpm for 10 min. Then, 300 µL of all treatment’s supernatant for triplicate were placed in a 96-well microplate. Optical density was taken at 540 nm in a microplate spectrophotometer (Multiskan Go, Thermo Scientific, Waltham, MA, USA). The results were reported as hemolysis percentage (HP) using Equation (7):HP = [(A_Sample_ − A_PBS_)/(A_Triton_ − A_PBS_)] × 100(7)

### 2.9. Evaluation of the Erythroprotective Potential

For the evaluation of the erythroprotective potential of the extracts (PS, T-HCl, and MetOH), oxidative, osmotic, and heat hemolysis inhibition assays were carried out. The results were reported as hemolysis inhibition percentage (HIP) with Equation (8):HIP = [(A_Control_ − A_Sample_)/(A_Control_)] × 100(8)

#### 2.9.1. Antihemolytic Activity in Erythrocytes by the 2,2′-azobis-(2-methylpropionamidine) (AAPH) Method

Oxidative hemolysis was induced with the AAPH compound, which appears as a free radical initiator, based on the methods proposed [49,50]. They incubated the negative control (300 µL of erythrocytes), positive control (150 µL of erythrocytes + 150 µL of AAPH), and sample (100 µL of erythrocytes + 100 µL of extract + 100 µL of AAPH) at 37 °C for 3 h (Boekel Scientific, model 132000). 1 mL of PBS was added to each sample and centrifuged at 1500 rpm for 10 min. All treatments were placed in a 96 well-microplate for triplicate, using the supernatant of the reaction. Optical density was measured at 540 nm in a microplate reader (Multiskan Go, Thermo Scientific, Waltham, MA, USA).

#### 2.9.2. Hypotonicity-Induced Hemolysis Assay

Hypotonicity-induced hemolysis was carried out according to the methodology, with some modification [51]. Treatments were incubated in a water bath at 37 °C for 30 min: sample (50 μL of RBC + 100 μL of extract + 100 μL of PBS + 200 μL of hyposaline solution), negative control (50 μL of RBC + 400 μL of PBS), positive control (50 μL of RBC + 200 μL of PBS + 200 μL of hyposaline solution), and standard control (50 μL of RBC + 100 μL of sodium diclofenac (SD, 1 mg·mL^−1^) + 100 μL of PBS + 200 μL of hyposaline solution). After incubation, 850 μL of PBS was added and centrifuged at 1500 rpm for 10 min. Later, 300 μL of recovered supernatant of each treatment was added in a microplate of 96-well to be triplicated. The absorbance was recorded at 560 nm.

#### 2.9.3. Heat-Induced Hemolysis Assay

Heat-induced hemolysis was carried out according to the methodology with some modification [51]. There was mixing of the treatments: sample (150 μL of RBC + 150 μL of extract), positive control (150 μL of RBC + 150 μL of PBS), negative control (150 μL of RBC + 150 μL of PBS), and standard control (150 μL of RBC + 150 μL of sodium diclofenac (SD, 1 mg·mL^−1^). All treatments, excepting the negative control, were incubated at 55 °C for 30 min in a water bath. After incubation, 1 mL of PBS was added to each sample and centrifuged at 1500 rpm for 10 min. The amount of 300 μL supernatant of each treatment was placed for triplication in a 96-well microplate. The absorbance was recorded at 560 nm in a microplate lector (Multiskan Go, Thermo Scientific, Waltham, MA, USA).

## 3. Results and Discussion

### 3.1. Growth Kinetics

The growth kinetics of *P. cruentum* showed a sigmoidal behavior, which can be seen in Figure 1. With the results of the average cell concentration, the day of cultivation was determined as the harvest point, which corresponds to the beginning of the phase stationary, with an average cell concentration of 2,358,899 ± 336,481 cells mL^−1^, which is equivalent to 21.16 ± 0.20 cells mL^−1^ in log_2_ (cell·mL^−1^).

Differences in culture conditions generate variability in growth kinetics and cell concentrations reported for *P. cruentum* (2,358,899 ± 336,481 cells mL^−1^). The results agree with another report, since they obtained an average cell concentration of 3,220,000 ± 280,000 cells mL^−1^ on day 8 of culture, which marked the beginning of the stationary phase [52]. Furthermore, day 10 of culture was defined as the starting point of the stationary phase of *P. purpureum* (2.3 g/L) [34], and data that support the growth kinetics are outlined in the current project.

The cultivation of the GUMACC 25 (UTEX 161) strain of *P. cruentum* was carried out in Sarstedt cell culture flasks with a photoperiod of 18:6 [52]. Due to the size of the microalgal cell (4 to 9 µm), its planktonic behavior, and its ability to form cell aggregates [53,54], Sarstedt cell culture flasks can favorably influence the increase in microalgal biomass by presenting a higher surface–volume ratio [55], otherwise, it has been proven that a constant photoperiod is usually optimal for the production of microalgal biomass [56].

In that same context, the light administered (98 μmol m^−2^ s^−1^) [52] was higher than the applicated in the current work (35 μmol m^−2^ s^−1^), therefore, it was verified that the own strain has a better response to lower irradiances. The information collected experimentally coincides with the data of different works [53,57], which recovered a higher amount of biomass and B-PE in the lower light intensity treatment of the species in *Porphyridium purpureum* (67.46 and 60 μmol·m^−2^·s^−1^, respectively).

In addition to the differences in culture conditions, there is also one in the estimation of cell concentration. It was carried it out by in vivo fluorescence of chlorophyll [52], and in the present work by manual counts with a Neubauer camera under the optical microscope. Indirect methods for determining the cell concentration of microalgae, such as the in vivo fluorescence method of chlorophyll, are simpler, but their main drawback is that they must be accurately calibrated [58]. However, the low concentration of chlorophyll α present in *P. cruentum* [59,60] could affect the weighting of the cell density of the species using this method, so traditional counts under the light microscope could be considered the best for the determination of the cellular concentration of *P. cruentum.*

### 3.2. Pigments and Protein Quantification

Phycobiliprotein concentrations were significantly different in the T-HCl extract; in the case of the PS extract, there were no significant differences between R-PC and APC. B-PE showed the highest concentration in both extracts. Likewise, the APC was the only PBP that did not show significant differences between the extracts (Table 3). On the other hand, concentrations of C_total_ and C_x+c_ were relatively low (Table 4).

In the present work, APC was successfully detected, a process that usually presents difficulties due to its low intracellular concentration and its strong and direct union with photosystem II through the core–membrane complex. There are three major proteins in the core–membrane complex of the phycobilisome: the core–membrane linker protein (LCM) and two APC variants (β18.5 and αAP-B). The LCM is involved in the assembly of the phycobilisome nucleus and the interactions between the nucleus and the thylakoid membrane [54]. The LCM and β18.5 proteins are responsible for the close interaction between the phycobilisome and photosystem II, which is probably necessary for efficient energy transfer from phycobilisome to photosystem II. The close association between phycobilisome and photosystem II is consistent with a recent report that demonstrated the formation of the phycobilisome-photosystems II-photosystem I megacomplex in vivo through protein crosslinking and mass spectrometry analysis [54,61]. This phenomenon, typical of the nature of *P. cruentum*, has consequences in the extraction of the APC due to the characteristic hydrophobicity of the LCM; in fact, it was not possible to detect the APC in studies of *P. cruentum* and *P. purpureum*, respectively [62,63]. Low concentrations of chlorophylls and total carotenoids are a common aspect in the species since energy transfer is carried out mainly by PBP [64,65].

PBP concentrations were higher in the present work than the reported for the same species [66]; this may be due to extraction, since methanol removes non-polar compounds, such as chlorophylls and carotenoids. In addition, the implosion of the created cavitation bubbles promotes the rupture of the outer membrane of the cells and facilitates the penetration of aqueous solutions to extract a greater amount of PBP, miscible by its hydrophilic nature [67].

In the current project, two aqueous solutions were used due to their different characteristics to extract the PBP: PS and T-HCl. The commercial PS is a substance for hospital use, approved by the FDA in its administration for the replacement of extracellular fluid (due to dehydration, hypovolemia, hemorrhage, sepsis, among others), the treatment of metabolic alkalosis, slight sodium reduction, as well as its use during common procedures, such as blood transfusions and hemodialysis [68]. Tris(hydroxymetal)aminomethane (T-HCl) buffer is one of the most widely used buffers in biochemistry and molecular biology due to its high stability and compatibility [69], particularly in in vitro studies [70]. The concentration of PBP was higher in the extract made with the T-HCl buffer, which may be a consequence of its composition, and the differences in its physicochemical characteristics such as salinity and pH are relevant during the extraction. PS consists of 154 mmol L^−1^ Na^+^ and 154 mmol L^−1^ Cl^−^, and 0.5 M T-HCl of 500 mmol L^−1^ NH_2_C(CH_2_OH)_3_·HCl, both dissolved in water [70,71]. Due to its lower ionic concentration, the T-HCl buffer could increase the PBP solubility in water and their ionic strength, avoiding protein precipitation and contaminating agents due to the salting out effect [72]. In addition, the low pH of 5.5 of the PS [73], compared to the pH of 6.5 of the T-HCl buffer, could cause an internal electrostatic attraction generating a net positive charge due to the donation of H^+^, promoting protein opening and release of the solvent, a phenomenon that could result in the denaturation of the PBP [72,74], which leads to a lower net concentration of PBP in the PS extract.

The protein concentration of the aqueous extracts is shown in Table 5, where the concentration of the PS extract was significantly higher than that of the T-HCl extract. Although total protein concentration of the PS extract was higher than T-HCl protein concentration, the latter one has a higher concentration of phycobiliproteins, so the proportion of proteins and phycobiliproteins in the extracts has a direct impact on the bioactive properties of both extracts.

### 3.3. Biological Activities of Phycobiliproteins Analysis

Based on the pigment’s quantification, the antioxidant activity of the MetOH extract could be influenced by the presence of chlorophylls, carotenoids, and, according to their polarity, by polyunsaturated and highly unsaturated fatty acids and exopolysaccharides. Likewise, the antioxidant activity of the aqueous extracts is supported by the high concentration of PBP, and tentatively by endopolysaccharides, due to their intracellular location and their hydrophilic nature [75,76].

#### 3.3.1. Antioxidant Activity

Antioxidant activity provided by the extracts was significantly different in each method (*p* < 0.001). The PS extract showed the highest antioxidant activity by the ABTS^+•^ method, while in T-HCl and MetOH extracts by DPPH^•^. Likewise, the MetOH extract was the one that obtained the highest antioxidant activity by FRAP. The results of the antioxidant activity are condensed in Table 6.

Pearson’s correlation coefficient of the results of the FRAP and DPPH^•^ tests was greater than 0.95 for the aqueous extracts (Figure 2). On the contrary, there is no correlation between the antioxidant mechanisms of the FRAP assay and ABTS^+•^ (Figure 3). The Pearson correlation coefficient of the results of the ABTS^+•^ and DPPH^•^ tests was greater than 0.95 in the MetOH and T-HCl extracts (Figure 4).

The ABTS^+•^, DPPH^•^, and FRAP assays make it possible to estimate the antioxidant capacity through different oxidation-reduction reactions. In addition, the three techniques executed are part of the guild of methods with the highest frequency of analysis of antioxidant activity of food, nutrition, and supplements [22]. The application of a variety of assays with different mechanisms (SET, HAT, and assays based on lipid peroxidation) lead to obtaining a more precise antioxidant activity profile [77]. With this premise, the performance of three different antioxidant tests qualifies the present work as appropriate; in fact, the selection of the ABTS^+•^, DPPH^•^, and FRAP assays is considered excellent for the evaluation of antioxidant capacity due to its complementarity [78].

The differences between the antioxidant capacity of the extracts depending on the methods carried out can be explained by the HAT and SET antioxidant mechanisms specific to each trial carried out [79,80]. PBP concentration in the T-HCl extract is higher than the PS extract. Because PBP are water-soluble proteins, they are incompatible with the organic solvent of the DPPH^•^ assay [77,80], although this disadvantage could have been displaced by the cysteine content in the PBP. Cysteine is the only amino acid detectable by DPPH^•^ [81], and it was possibly a key factor in achieving a high antioxidant activity of the T-HCl extract. In relation to this, a higher concentration of proteins in the PS extract may have an impact on its antioxidant activity. In case these proteins are hydrophilic, they could interact with the PBPs, promoting their precipitation in the alcoholic environment of the bioassay [77,82,83].

Sulfated endopolysaccharides represent other presumed bioactive compounds of *P. cruentum* involved in the antioxidant activity of the aqueous extracts, since a preliminary organic extraction is required for their extraction and purification, which could be covered with the methanolic extraction of the present study after acid extraction, as performed by PS (pH 5.5) and, to a lesser extent, by T-HCl (pH 6.5). Consequently, the concentration of hydrophilic endopolysaccharides could be higher in the PS extract, the effect of which could go unnoticed due to the hydrophobicity of the DPPH^•^ assay [75,78]. However, if these act through the HAT mechanism, their antioxidant effect could not have been detected by FRAP either. In retrospect, the hydrophobicity of the DPPH^•^ assay and the restriction of the SET antioxidant mechanism in the FRAP assay could affect the determination of the antioxidant activity of the likely-sulfated endopolysaccharides traces in the PS and T-HCl extract, the impact of which is visualized in a lower antioxidant activity in the FRAP assay compared to the MetOH extract.

The significantly higher antioxidant activity performed by the MetOH extract in the DPPH^•^ and FRAP assays indicates that it has antioxidant compounds that operate by SET mechanism, surpassing the hydrophilicity of the FRAP assay. In this sense, the electron transfer capacity does not represent the predominant antioxidant mechanism of the aqueous extracts. This can be confirmed by the results of the ABTS^+•^ assay, where the antioxidant activity of the aqueous extracts was significantly higher than that of the MetOH extract, so it can be deduced that they act mainly through the HAT antioxidant mechanism.

Generally, SET-based assays correlate well with each other. Although the intervening compounds are different, the basis of the action mechanism is similar, which is why the application of a series of SET-based assays used to be classified as redundant. On the other hand, the assays based on HAT and SET are not necessarily correlated [22], as was the case with all the extracts in the ABTS^+•^ and FRAP assays, which function under different polarity and antioxidant mechanisms.

Strong correlations between assays demonstrate their effectiveness and complementarity to the antioxidant activity evaluation [84]. In this case, a strong correlation was determined in the results of the antioxidant activity of the FRAP and DPPH^•^ assays, except for the MetOH extract, which can be supported by a significantly higher antioxidant activity, promoted by the prevailing SET mechanism; similarly, a strong correlation was obtained between the antioxidant activity results of the DPPH^•^ and ABTS^+•^ assays, but not in the PS extract, where the HAT antioxidant mechanism could be even more predominant than in the T-HCl extract. The differences in the concentrations of the bioactive compounds of the PS extract affected the correlation of the DPPH^•^ and ABTS^+•^ results, which could be attributed to a lower concentration of PBP that acts mainly by HAT, a higher concentration of other proteins that affect the availability of PBP as antioxidants by interacting with them, and a higher concentration of water-soluble endopolysaccharides that could promote the antioxidant activity in the ABTS^+•^ assay but not the DPPH^•^ assay because of the exclusive affinity for hydrophobic antioxidant compounds. It is suggested that the aqueous extracts carry out both antioxidant mechanisms, mainly the HAT, whose predominance is even greater in the PS extract.

The ability to sequester ROS, attributed to carotenes, and that of peroxyl sequestration, provided by xanthophylls, as well as the electron transfer conferred by chlorophylls, together to a lesser extent with the HAT mechanism [85], are involved in the antioxidant activity of the methanolic extract, carried out mainly by SET, a phenomenon revealed by the Pearson correlation analysis of the FRAP, DPPH^•^, and ABTS^+•^ assays in such a way that the results obtained have biochemical support and statistics.

The redox and metal-chelating properties of phycobiliproteins, particularly their chromatophores, have been demonstrated by various antioxidant activity assays: ORAC, TRAP, β-carotene or crocin bleaching, FRAP, lipid peroxidation inhibition, TBARS, CUPRAC, DPPH^•^, ABTS^+•^, and HORAC, among others [86]. PBP inhibits hydroxyl radicals, and this capacity is directly proportional to the content of phycobilins, which act as ROS scavengers [85].

Each phycobiliprotein has mechanisms of antioxidant activity. It has been established that PE act through the primary route, through the direct sequestration of ROS, and they also have a high reducing power. On the contrary, the PC and APC carry out both routes, primary and secondary, standing out as chelators of metal ions that participate in the synthesis of ROS [87,88]. The antioxidant mechanisms also depend on the variations in the distribution of amino acids. Among them, they stand out for their content of hydrophobic aminoacids, considered excellent proton donors and metal ion chelators, as well as acid, basic, and aromatic aminoacids, which are metal ion sequestrants [89]. In this sense, the high reducing power carried out mainly by HAT of the aqueous extracts demonstrated in the antioxidant tests carried out is due to the significantly higher concentration of PE compared to that of PC and APC. In addition, it has been found that the HAT antioxidant mechanism is dominant in aqueous solutions, such as aqueous extracts; likewise, the SET mechanism predominates in organic solvents, where hydrogen bonds can be created with antioxidant molecules [90,91,92,93].

#### 3.3.2. Blood Biocompatibility Assay

Blood biocompatibility results of the aqueous and methanolic extracts on ABO antigens and A, O, and AB groups with RhD positive and negative are shown in Figure 5, Figure 6, Figure 7 and Figure 8, respectively. The evaluation of the results of the blood biocompatibility test is based on the critical limit of hemolysis, which must be less than 5% (ISO/TR 7406). With this base, it was determined that the aqueous extracts are innocuous on the A Rh+ve, B Rh+ve, and O Rh+ve groups, since they did not exceed the critical limit of hemolysis, with the exception of the PS extract on the B Rh+ve group and the two aqueous extracts on the AB Rh+ve group, however, the damage developed is considered minimal and stable because it did not increase between hours 4 and 6 of exposure.

The results of the blood biocompatibility test of all the extracts on groups A, O, and AB with RhD positive and negative outline different trends. All the extracts were innocuous on the A antigen. The absence of the RhD, or the negative RhD of group O, seems to affect the blood biocompatibility of the PS extract, but the percentage of hemolysis did not increase between hours 4 and 6 (Figure 7), which was another explanation. An alternative is that the presence of the RhD or a positive RhD is what promotes the erythroprotective effect of the extract. The critical limit of hemolysis was exceeded in the AB Rh+ve group in the PS and T-HCl extract evaluation, although this did not show significant differences between hours 4 and 6 and between hours 2 and 6 of exposure, respectively. Therefore, group B and/or the presence of the RhD could affect the protection of the erythrocyte membrane.

The blood biocompatibility of the methanolic extract was only suitable on the A antigen and the AB Rh+ve group, where the percentage of hemolysis was less than 5%. The antigen that presented a higher percentage of hemolysis due to its interaction with the methanolic extract was B. Despite this, it did not increase between hours 2 and 6 (Figure 5). The percentage of hemolysis detected in the methanolic extract increased during the period of the test in the O group, while this was maintained between 4 and 6 h in the AB Rh+ve group, both in the positive and negative RhD.

The hemolysis promoted by the methanolic extract in the erythrocytes of different blood groups is tied to methanol, which is a solvent capable of reducing the mechanical stability of the cell, as well as the fluidity of the hydrophobic nucleus and of the membrane near the surface. In addition to promoting the deformability of RBC [94], although the results indicate that hemolytic damage is stabilized, for good cell preservation it is recommended to remove methanol by rotoevaporation or other methods, with the purpose of taking full advantage of the bioactive compounds contained in the extract.

The rest of the results with a percentage of hemolysis inhibition greater than 5% seem to be influenced by the specific interactions between the group antigens and the extract (on PS for the presence of B antigen in group B Rh+ve; on T-HCl and PS for the presence of the B antigen in the AB Rh+ve group) or between the RhD and the extract (PS due to the absent RhD on O group), while in the case of MetOH, it was due to the nature of the extract. Within this framework, the evaluation of the cytotoxic action of the microalgae extracts on each blood group was elemental.

In a general way, it was evidenced that the aqueous and methanolic extracts from *P. cruentum* present competent blood biocompatibility, so the performance of other bioassays aimed at evaluating different biological activities on human erythrocytes is appropriate. In the present work, these a posteriori tests consisted of the evaluation of their erythroprotective potential.

#### 3.3.3. Erythroprotective Potential

The antioxidant mechanisms of the extracts, mainly HAT in the PS and T-HCl extracts, and SET in the MetOH extract, regulate the erythroprotective effect against radicals generated by the AAPH compound, hypotonicity, and heat, preventing oxidative hemolysis. The inhibition of the hemolysis accomplished by the *P. cruentum* compounds in the extracts could be carried out even before their contact with the plasmatic membrane, through an interaction with the erythrocyte membrane surface glycoproteins and proteins, depending on the Rh or ABO system group, which could imply the formation of a protective complex against the oxidation of the membrane promoted by free radicals [77] that have been produced by the AAPH agent, hypotonicity, or heat. Another alternative is that the extracts could modify the erythrocyte membrane by increasing the surface area/volume ratio, which results in membrane expansion or cell shrinkage, changing the interaction with membrane proteins and protecting the cell against hemolysis [95].

The nature of the AAPH-induced hemolysis, hypotonicity, and heat assays underpin the difference between their results. The AAPH is an initiator of alkoxyl and peroxyl radicals, pointed out as the main generators of damage to the erythrocyte membrane. Therefore, the extracts provided protection against oxidative damage. On the other hand, due to their absence of organelles, erythrocytes are considered analogous to lysosomes. When erythrocytes are subjected to a hypotonic medium, they respond with adaptations in osmosis and cell tonicity, an action that defined the stability of the erythrocyte in the face of osmotic changes, and, consequently, that of the lysosome, whose stability in the face of osmotic changes is transcendental due to its high content of reactive species and proteolytic enzymes, important in inflammatory responses, and, in parallel, in the stability of erythrocytes. Likewise, temperature generates changes in the internal cell environment, including lysosomes, where temperature-dependent enzymatic reactions are generated, so they can exceed basal temperature ranges, which justifies the determination of their stability according to the temperature [96,97].

The differential susceptibility to oxidative damage on the erythrocyte plasma membrane could be associated with the blood groups of the ABO and RhD system to which they belong. In addition, there are numerous association studies between diseases and blood groups. Additionally, in the present work it was observed that the degree of the microalga extract’s biological activities could depend on the membrane antigens. The importance of these phenomena elucidates the creation of biotargeted drugs to a certain blood group, to treat and/or prevent communicable and non-communicable diseases efficiently, in favor of human health.

##### Antihemolytic Activity in Erythrocytes by the 2,2′-azobis-(2-methylpropionamidine) (AAPH) Method

Microalgal extracts were able to differentially inhibit the reactive species synthesized from AAPH, depending on the blood groups of the ABO and RhD system. In group A, microalgal extracts provided a higher percentage of hemolysis inhibition, exceeding that provided by the standard antioxidants AA, BTC, and FXA. On the other hand, in groups B and AB, the inhibition of hemolysis resulting from the aqueous extracts was significantly less. On the other hand, the MetOH extract carried out the highest antihemolytic activity in all blood groups (Figure 9).

The evaluation of the influence of RhD on the antihemolytic capacity of the extracts indicates that the aqueous extracts had a greater erythroprotective effect in the A Rh+ve group compared to the A Rh-ve group (Figure 10). Likewise, the MetOH extract generated the greatest erythroprotective effect in the O Rh+ve group and that of T-HCl in the O-, both equivalent to that of FXA (Figure 11). In the case of the AB Rh+ve group, the MetOH extract performed a greater antihemolytic function than the aqueous extracts, equivalent to that of the standard antioxidants, with more favorable results in the AB Rh+ve group (Figure 12).

AAPH is a free radical initiator capable of promoting oxidative hemolysis through lipid peroxidation and oxidation of erythrocyte membrane proteins. The accumulation of oxidized lipids around band 3 possibly leads to the formation of holes in the erythrocyte membrane, leading to hemolysis [6,98]. The ROS generated by the AAPH compound, especially H_2_O_2_, a physiological cellular metabolite produced in enzymatic and non-enzymatic reactions, can trigger the oxidation of membrane proteins, such as hemoglobin, and of lipids, particularly cholesterol, promoting damage in cell morphology and membrane structure, and ultimately hemolysis [84]. HAT is considered the antioxidant mechanism of the tests based on the AAPH compound [99], which supports the outstanding antihemolytic capacity of the aqueous extracts, which primarily carry out the HAT antioxidant mechanism, while in the MetOH extract, despite the fact that the HAT mechanism does not represent its main antioxidant mechanism, is still considered remarkable, and was effective in overcoming its prohemolytic activity because the methanol in the presence of free radicals was produced by the AAPH compound.

Extracts of *P. cruentum* exerted a protective effect on erythrocytes against oxidative hemolysis generated by the AAPH compound, with different results depending on the blood group system. The aqueous extracts performed a total inhibition of hemolysis on phenotype A. In group O, a total inhibition of hemolysis was also generated with the PS extract, but with less erythroprotection exerted by the T-HCl extract, with a reduced antihemolytic effect in groups B and AB effected by both aqueous extracts. This could denote that the B antigen, present in groups B and AB, affects the erythroprotective effect of the aqueous extracts, and in addition, the possibility that the A antigen increases its antihemolytic activity. This is relevant, since group A is the one with the highest incidence of COVID-19 and certain carcinomas [9,10,11].

The MetOH extract carried out an absolute erythroprotective effect in all blood groups. The MetOH extract is usually discarded in the PBP extraction process [100,101], but due to its potential as an antihemolytic agent, it can be considered as a by-product.

The results indicate that the presence or absence of RhD has an influence on the erythroprotective effect affected by the extracts, to a greater or lesser extent depending on the group of the ABO system to which it belongs. In retrospect, the presence of the RhD leads to better erythroprotection in group A by the aqueous extracts. On the other hand, the RhD did not affect the erythroprotective capacity promoted by the MetOH extract. In group O, the MetOH extract had a better erythroprotective effect with the presence of RhD, while the T-HCl extract showed a higher antihemolytic capacity due to the absence of RhD. Similarly, the aqueous extracts had a minor erythroprotective effect on group AB, but the MetOH extract stood out for its antihemolytic capacity equivalent to that of standard antioxidants, in which the presence of RhD allowed greater protection of the erythrocyte membrane.

Investigations of the erythroprotective effect of different microalgal compounds against AAPH-induced hemolysis are limited. The antihemolytic capacity of a 0.75 mM PC sample was determined to be 97%, however, the blood group of the experiment was not specified [31]. Additionally, it was evaluated the erythroprotective effect of extracts from the *Navicula incerta* and obtained high percentages of hemolysis inhibition with an acetone (89.20% ± 0.70), methanolic (96.70% ± 1.10), and ethanolic (81.80% ± 3.20) extract on group O [102]. Based on the same type of extract and blood group, in the present work, the percentage of inhibition of hemolysis detected was higher (100.00% ± 0.41). On the other hand, antihemolytic activity of extracts has been reported, originating from *N. incerta* with a percentage of hemolysis inhibition affected by the methanolic extract for group A (68.67% ± 5.36), B (76.33% ± 3.51), and O (77.73% ± 4.03), also considering the RhD, with a percentage of hemolysis inhibition for group O RhD positive (69.09% ± 2.44) and RhD negative (68.16% ± 3.21) [103]. Making a comparison, the results of the present study indicate a greater erythroprotective capacity of the MetOH extract for all groups: A (98.58% ± 1.74), B (100.00% ± 0.41), and O (100.00% ± 0.41).

##### Hypotonicity Induced-Hemolysis Inhibition

The results of the hypotonicity-induced hemolysis test on ABO antigens had a similar behavior in the different blood groups (Figure 13). In this way, the blood group that developed a greater erythroprotective protection due to the action of all the extracts was O, followed by B and A, while the AB group showed a variable behavior, surpassing A only in the treatment of the T-HCl extract and SD, but the effect was less compared to the rest of the groups with each of the extracts.

The inhibition of hemolysis induced by hypotonicity on the RhD was complementary, since there are significant differences in the results corresponding to the positive and negative RhD of all the groups evaluated. In such a way that in the A Rh+ve group the aqueous extracts had a greater inhibitory effect on hemolysis than in the A Rh-ve group, which benefited more from the MetOH extract (Figure 14). In the O group, the inhibition of the hemolysis of the aqueous extracts was independent, while a greater protection of the erythrocytes was developed in the O Rh+ve group with the T-HCl extract. In the O Rh-ve group it was with the PS and MetOH (Figure 15). The AB Rh-ve group showed a higher percentage of hemolysis inhibition with all the extracts and with the SD compared to the AB Rh+ve (Figure 16).

The evaluation of the influence of RhD on the antihemolytic capacity of the extracts indicates that the aqueous extracts had a greater erythroprotective effect in the A Rh+ve group compared to the A Rh-ve group (Figure 14). Likewise, the MetOH extract generated the greatest erythroprotective effect in the O Rh+ve group and that of T-HCl in the O-, both equivalent to that of FXA (Figure 15). In the case of the AB Rh+ve group, the MetOH extract performed a greater antihemolytic function than the aqueous extracts, equivalent to that of the standard antioxidants, with more favorable results in the AB Rh+ve group (Figure 16).

The stability of the erythrocyte membrane can be extrapolated to the stabilization of the lysosomal membrane, due to its absence of organelles and similarity to the cell membrane [104]. During the inflammatory process, leukocytes and neutrophils release lysosomal enzymes towards the inflammation sites [105,106], and these can generate persistent inflammation and tissue damage, threatening human health. The osmosis effect of vascularization during inflammation is reproduced by the hypotonic solution, an analysis where the stabilization of the RBC membranes represents the lysosomal membranes, whose importance lies in limiting the inflammatory response by preventing the release of lysosomal enzymes. Therefore, the induction of hemolysis induced by hypotonicity is a method widely used in research as a biochemical parameter for tests of anti-inflammatory activity in vitro [107].

The hypotonic solution used in the assay promotes hemolysis due to excessive accumulation of fluid in the cells, causing damage to the cell membrane, which increases its susceptibility to secondary damage. Secondary damage refers to lipid peroxidation induced by free radicals [108,109], which can be prevented by membrane stabilization, an action possibly effected by aqueous microalgae extracts and MetOH from *P. cruentum*.

The results of the hypotonicity-induced hemolysis test on the ABO antigens had a similar behavior in the different blood groups, where a greater antihemolytic activity was observed by all the extracts in group O, followed by B, A, and finally AB. These results suggest that the presence of A and B antigens could affect the inhibition of hemolysis in the face of osmotic changes in the extracts. There is a probable association of the RhD and the protection of the extracts against hemolysis induced by hypotonicity, since it was observed that the microalgal extracts and the SD had a membrane cell better protection in the absence of the RhD, except for the MetOH extract in A group and T-HCl in O.

In a study, it was determined the inhibitory capacity of hemolysis induced by hypotonicity of microalgae extracts from different species. Although the blood group was not specified, comparisons were made with the O Rh+ve group due to its population frequency. They obtained high percentages of inhibition of hemolysis, from a methanolic extract of *Padina pavonica* (77.20 ± 0.24), methanolic extract from *Jania rubens* (79.08 ± 0.02), ethanolic extract from *Taonia atomaria* (57.50 ± 0.23), and methanolic extract from *Corallina elongata* (66.70 ± 0.05) [110], which, in contrast to the results of the extracts of *P. cruentum* of T-HCl (100.00 ± 3.46) and MetOH (87.30 ± 4.07), were higher, except for PS (74.87 ± 9.13), which can be explained by a lower concentration of B-PE, R-PC, and net PBP compared to the T-HCl extract, which represents its main bioactive compound content.

##### Heat Induced-Hemolysis Inhibition

The inhibition of heat-induced hemolysis by the different microalgae extracts was heterogeneous in all blood groups (Figure 17, Figure 18, Figure 19 and Figure 20). The aqueous extracts were responsible for less than 40% inhibition of hemolysis in all blood groups, with lower results than in the rest of the trials. The PS extract performed greater erythroprotective protection in groups A and O, and in a complementary way, the T-HCl extract in group B, although it did not prevent hemolysis in group AB. However, this group was the one that developed a greater inhibitory capacity of hemolysis with the MetOH extract, to which the best results of the present assay are attributed, superior to that of the SD.

Considering the RhD, the results that indicated a greater inhibitory capacity of hemolysis were those carried out by the MetOH extract for groups A Rh+ve, A Rh-ve, AB Rh+ve, AB Rh-ve, and O Rh+ve. In contrast, in the O Rh-ve group, no significant differences were detected in the percentage of inhibition of hemolysis triggered by the MetOH extract and the T-HCl extract (Figure 19).

Inflammation promotes lysis of RBC, which causes a loss of cellular activity and a release of intracellular components. In addition, inflammatory mediators, such as cytokines, increase the permeability of the membrane, increasing its vulnerability to oxidative damage. The inhibition of RBC membrane destabilization by heat justifies its anti-inflammatory properties [111].

The MetOH extract stood out for its high inhibitory capacity of heat-induced hemolysis on ABO antigens. In fact, they were superior to that of SD, although the aqueous extracts also provided a lower hemolysis inhibitory capacity. In the case of group O, the presence of RhD potentiated the antihemolytic effect of the microalgal extracts, while the opposite occurred in group A, except with the PS extract (Figure 17). Apparently, the presence of RhD in the AB group annulled the hemolysis inhibitory capacity of the aqueous extracts, but not of the MetOH extract, even surpassing the protection provided to the AB Rh-ve group.

This was carried out a study in which they evaluated the erythroprotective capacity of extracts of the green alga *Enteromorpha intestinalis* through heat-induced hemolysis. They determined the percentages of hemolysis inhibition of methanolic (10.81 ± 01.12), ethanolic (11.21 ± 01.76), and hexanoic (13.09 ± 01.98) extracts [112]. These, in contrast to those of *P. cruentum* in the present study on group O: PS (25.08 ± 03.14), T-HCl (100.00 ± 3.46) MetOH (87.30 ± 4.07), were lower. The differences are attributed to the characteristics of each extract, such as the species of origin, the medium, and the compounds. *P. cruentum* extracts proved to be excellent anti-inflammatory and antihemolytic candidates.

## 4. Conclusions

Ultrasound-assisted extraction promotes the external membrane rupture of *P. cruentum*, allowing for better phycobiliprotein extraction. The antioxidant capacity of phycobiliproteins is supported by their antiradical potential (ABTS^+•^ and DPPH^•^) and reducing potential (FRAP). The antioxidant mechanisms of the extracts, mainly HAT in the PS and T-HCl extracts, and SET in the MetOH extract, regulate the erythroprotective effect against radicals generated by the AAPH compound, hypotonicity and heat, preventing oxidative hemolysis. The evaluation of the *P. cruentum* extracts erythroprotective potential revealed a probable association between the different antigens (ABO and RhD) with an antihemolytic effect of the phycobiliproteins. These phytochemicals could be the key to the development of drugs biotargeted to a certain blood group, and to treat and/or prevent communicable and non-communicable diseases efficiently in favor of human health.

## Figures and Tables

**Figure 1 metabolites-13-00366-f001:**
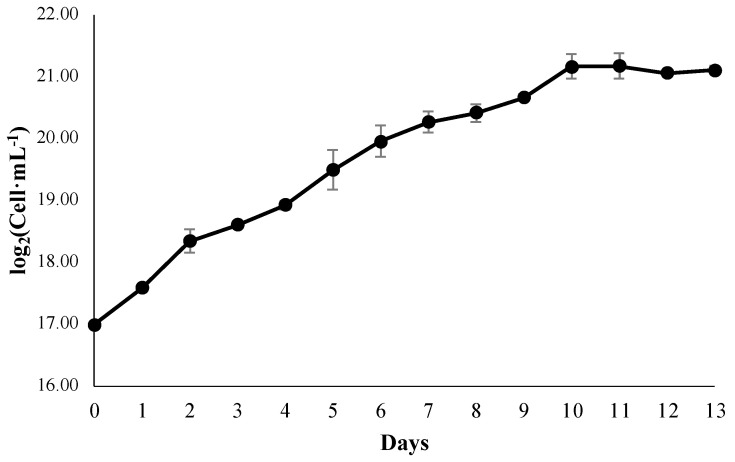
Growth kinetics of *P. cruentum* (log_2_ (cell·mL^−1^)).

**Figure 2 metabolites-13-00366-f002:**
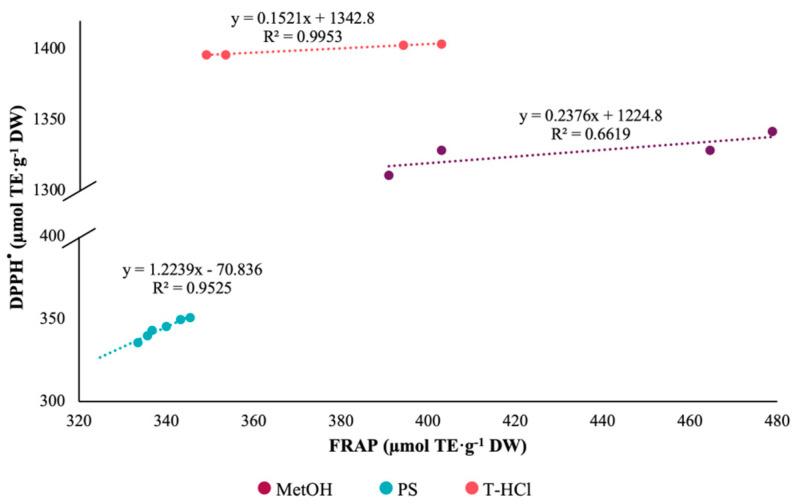
Pearson’s correlation of the FRAP and DPPH^•^ assays of MetOH, PS, and T-HCl extracts.

**Figure 3 metabolites-13-00366-f003:**
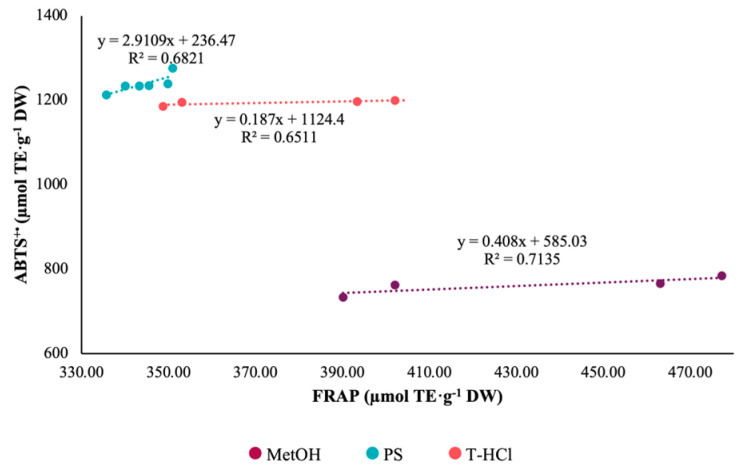
Pearson’s correlation of FRAP and ABTS^+•^ assays of MetOH, PS, and T-HCl extracts.

**Figure 4 metabolites-13-00366-f004:**
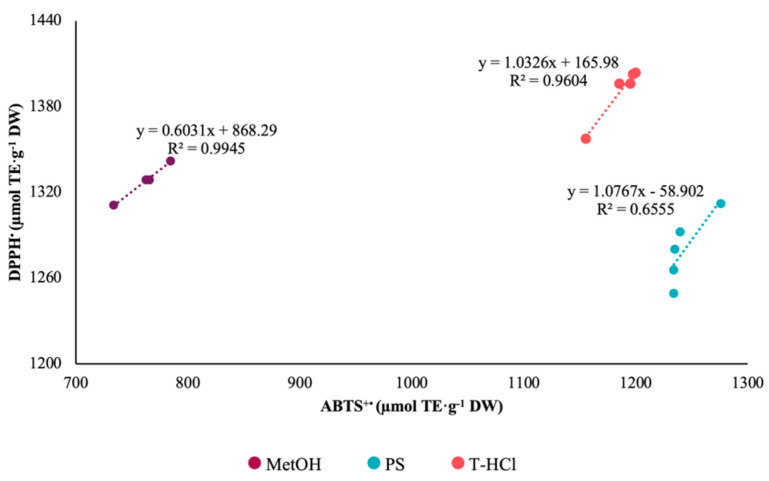
Pearson’s correlation of DPPH^•^ and ABTS^+•^ assays of MetOH, PS, and T-HCl extracts.

**Figure 5 metabolites-13-00366-f005:**
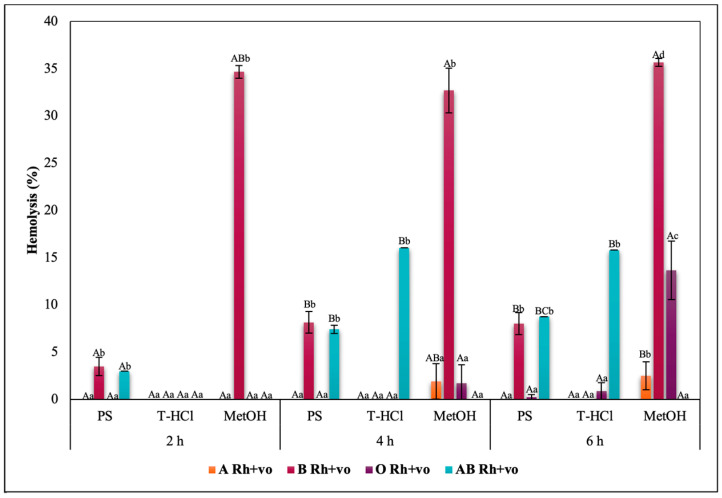
Blood biocompatibility assay of *P. cruentum* extracts on human erythrocytes with different ABO blood groups. Different letters represent significant differences in the means between treatments and groups (*p* ≤ 0.001). Capital letters represent a two-way ANOVA. Lowercase letters represent a univariate ANOVA between extracts for each exposure period. PS, extract of physiological solution; T-HCl, 0.5 M Tris-HCl extract; MetOH, 99% methanol extract; AA, ascorbic acid; BCT, beta-carotene; FXA, fucoxanthin.

**Figure 6 metabolites-13-00366-f006:**
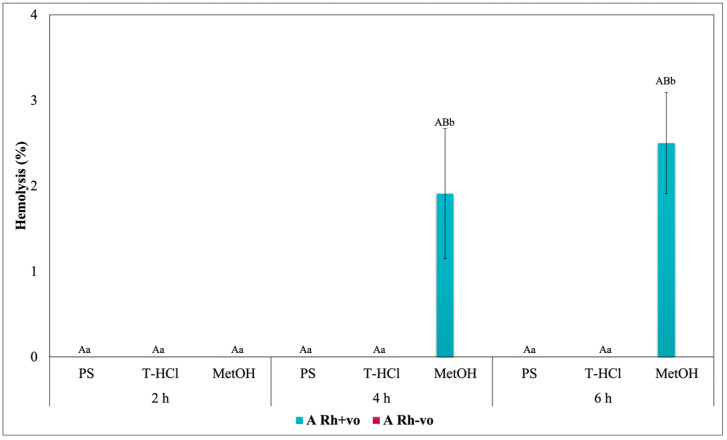
Blood biocompatibility assay of *P. cruentum* extracts on human erythrocytes with different RhD (positive and negative) of A blood group. Different letters represent significant differences in the means between treatments and groups (*p* ≤ 0.001). Capital letters represent a two-way ANOVA. Lowercase letters represent a univariate ANOVA between extracts for each exposure period. PS, extract of physiological solution; T-HCl, 0.5 M Tris-HCl extract; MetOH, 99% methanol extract; AA, ascorbic acid; BCT, beta-carotene; FXA, fucoxanthin.

**Figure 7 metabolites-13-00366-f007:**
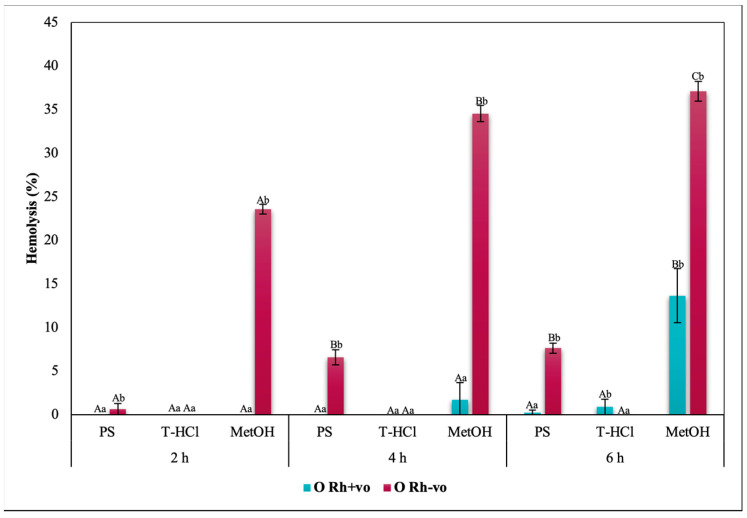
Blood biocompatibility assay of *P. cruentum* extracts on human erythrocytes with different RhD (positive and negative) of O blood group. Different letters represent significant differences in the means between treatments and groups (*p* ≤ 0.001). Capital letters represent a two-way ANOVA. Lowercase letters represent a univariate ANOVA between extracts for each exposure period. PS, extract of physiological solution; T-HCl, 0.5 M Tris-HCl extract; MetOH, 99% methanol extract; AA, ascorbic acid; BCT, beta-carotene; FXA, fucoxanthin.

**Figure 8 metabolites-13-00366-f008:**
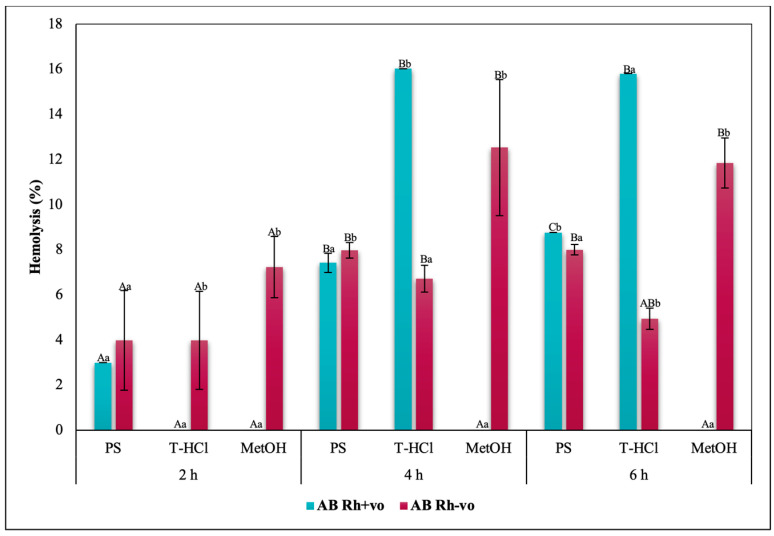
Blood biocompatibility assay of *P. cruentum* extracts on human erythrocytes with different RhD (positive and negative) of AB blood group. Different letters represent significant differences in the means between treatments and groups (*p* ≤ 0.001). Capital letters represent a two-way ANOVA. Lowercase letters represent a univariate ANOVA between extracts for each exposure period. PS, extract of physiological solution; T-HCl, 0.5 M Tris-HCl extract; MetOH, 99% methanol extract; AA, ascorbic acid; BCT, beta-carotene; FXA, fucoxanthin.

**Figure 9 metabolites-13-00366-f009:**
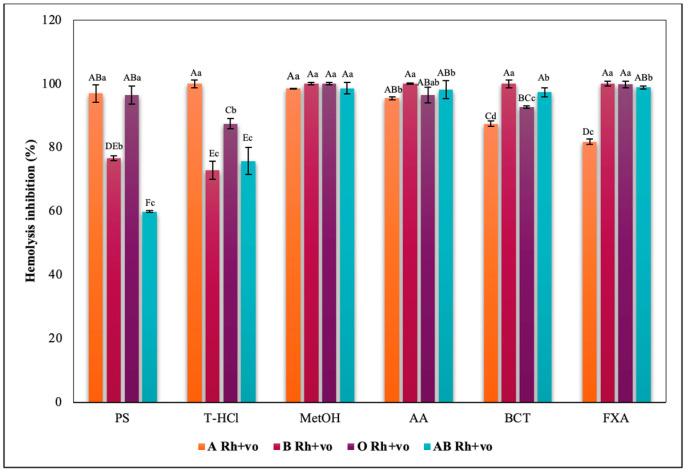
Inhibitory effect of hypotonicity-induced hemolysis of *P. cruentum* extracts on human erythrocytes with different ABO blood groups. Different letters represent significant differences in the means between treatments and groups (*p* ≤ 0.001). Capital letters represent a two-way ANOVA. Lowercase letters represent a univariate ANOVA between the extracts. PS, extract of physiological solution; T-HCl, 0.5 M Tris-HCl extract; MetOH, 99% methanol extract.

**Figure 10 metabolites-13-00366-f010:**
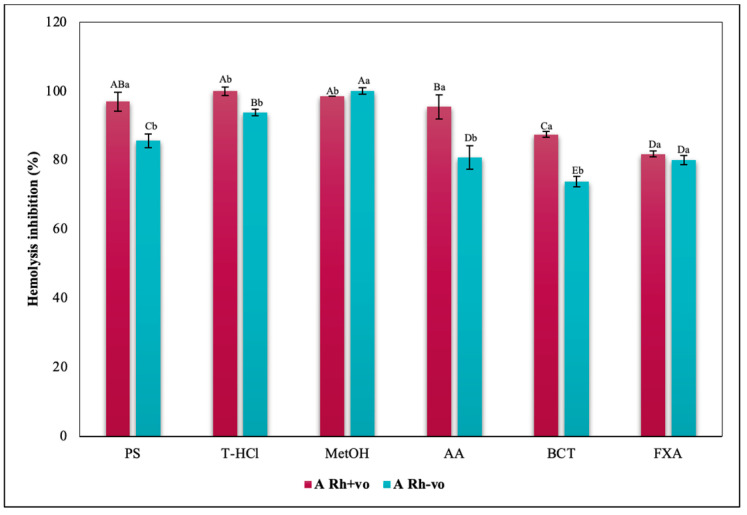
Inhibitory effect of hypotonicity-induced hemolysis of *P. cruentum* extracts on human erythrocytes with different RhD (positive and negative) of A blood group. Different letters represent significant differences in the means between treatments and groups (*p* ≤ 0.001). Capital letters represent a two-way ANOVA. Lowercase letters represent a univariate ANOVA between the extracts. PS, extract of physiological solution; T-HCl, 0.5 M Tris-HCl extract; MetOH, 99% methanol extract.

**Figure 11 metabolites-13-00366-f011:**
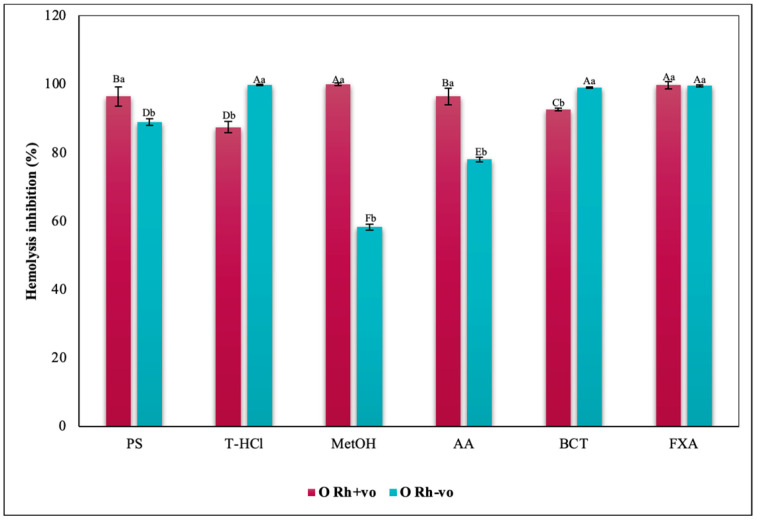
Inhibitory effect of hypotonicity-induced hemolysis of *P. cruentum* extracts on human erythrocytes with different RhD (positive and negative) of O blood group. Different letters represent significant differences in the means between treatments and groups (*p* ≤ 0.001). Capital letters represent a two-way ANOVA. Lowercase letters represent a univariate ANOVA between the extracts. PS, extract of physiological solution; T-HCl, 0.5 M Tris-HCl extract; MetOH, 99% methanol extract.

**Figure 12 metabolites-13-00366-f012:**
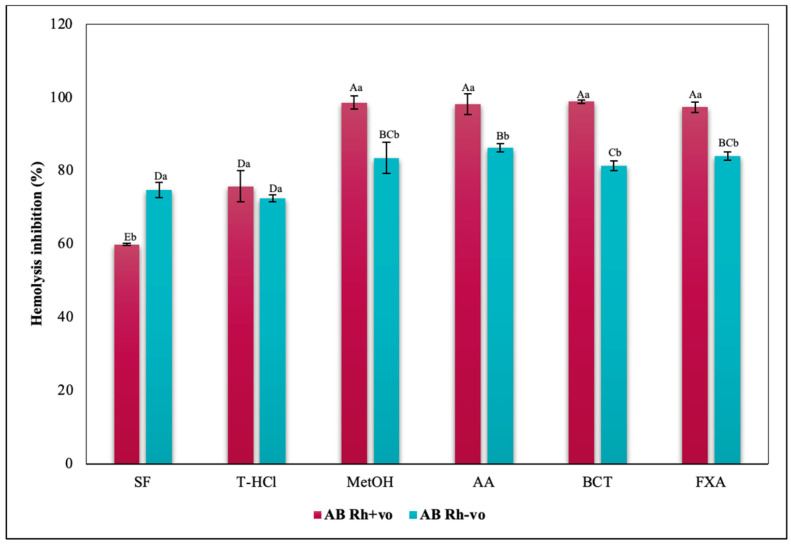
Inhibitory effect of hypotonicity-induced hemolysis of *P. cruentum* extracts on human erythrocytes with different RhD (positive and negative) of AB blood group. Different letters represent significant differences in the means between treatments and groups (*p* ≤ 0.001). Capital letters represent a two-way ANOVA. Lowercase letters represent a univariate ANOVA between the extracts. PS, extract of physiological solution; T-HCl, 0.5 M Tris-HCl extract; MetOH, 99% methanol extract.

**Figure 13 metabolites-13-00366-f013:**
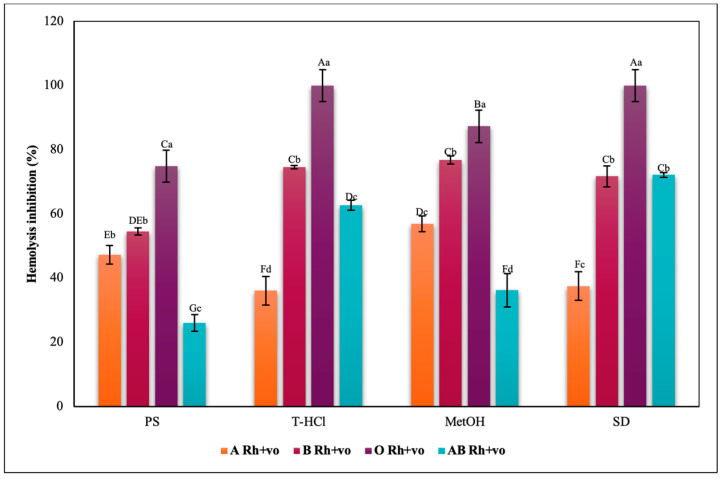
Inhibitory effect of hypotonicity-induced hemolysis of *P. cruentum* extracts on human erythrocytes with different ABO blood groups. Different letters represent significant differences in the means between treatments and groups (*p* ≤ 0.001). Capital letters represent a two-way ANOVA. Lowercase letters represent a univariate ANOVA between the extracts. PS, extract of physiological solution; T-HCl, 0.5 M Tris-HCl extract; MetOH, 99% methanol extract; SD, sodium diclofenac.

**Figure 14 metabolites-13-00366-f014:**
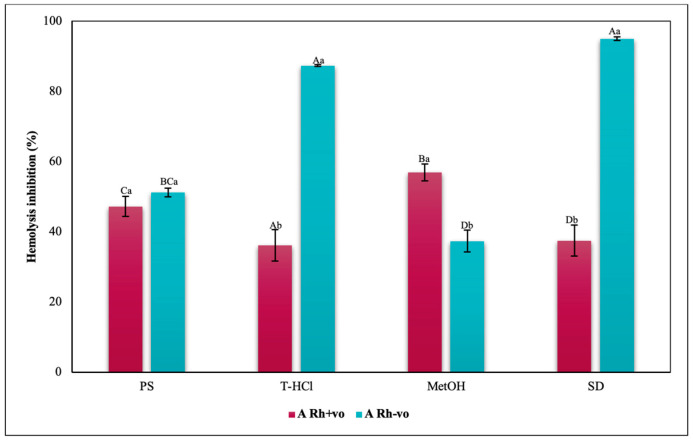
Inhibitory effect of hypotonicity-induced hemolysis of *P. cruentum* extracts on human erythrocytes with different RhD (positive and negative) of A blood group. Different letters represent significant differences in the means between treatments and groups (*p* ≤ 0.001). Capital letters represent a two-way ANOVA. Lowercase letters represent a univariate ANOVA between the extracts. PS, extract of physiological solution; T-HCl, 0.5 M Tris-HCl extract; MetOH, 99% methanol extract; SD, sodium diclofenac.

**Figure 15 metabolites-13-00366-f015:**
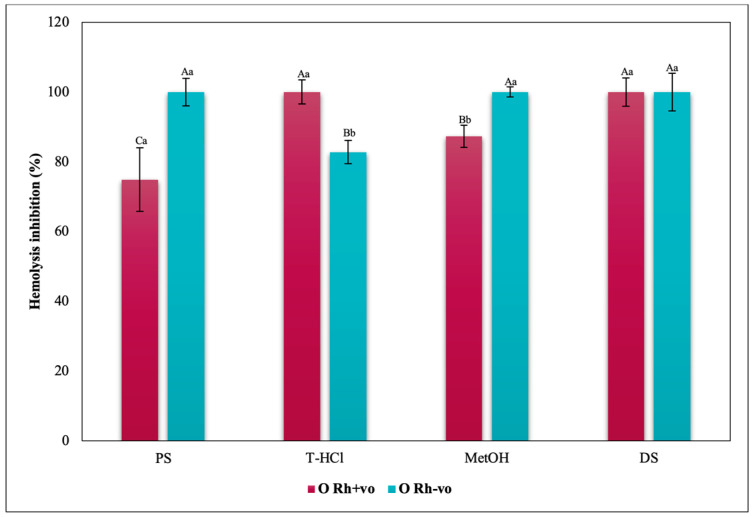
Inhibitory effect of hypotonicity-induced hemolysis of *P. cruentum* extracts on human erythrocytes with different RhD (positive and negative) of O blood group. Different letters represent significant differences in the means between treatments and groups (*p* ≤ 0.001). Capital letters represent a two-way ANOVA. Lowercase letters represent a univariate ANOVA between the extracts. PS, extract of physiological solution; T-HCl, 0.5 M Tris-HCl extract; MetOH, 99% methanol extract; SD, sodium diclofenac.

**Figure 16 metabolites-13-00366-f016:**
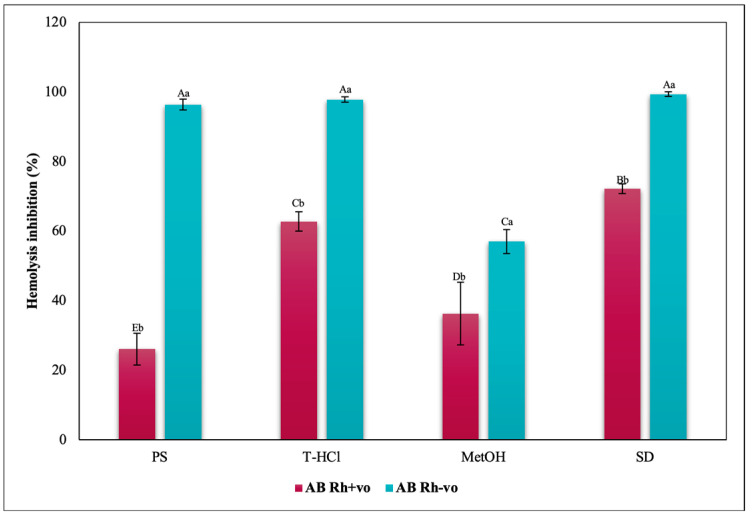
Inhibitory effect of hypotonicity-induced hemolysis of *P. cruentum* extracts on human erythrocytes with different RhD (positive and negative) of AB blood group. Different letters represent significant differences in the means between treatments and groups (*p* ≤ 0.001). Capital letters represent a two-way ANOVA. Lowercase letters represent a univariate ANOVA between the extracts. PS, extract of physiological solution; T-HCl, 0.5 M Tris-HCl extract; MetOH, 99% methanol extract; SD, sodium diclofenac.

**Figure 17 metabolites-13-00366-f017:**
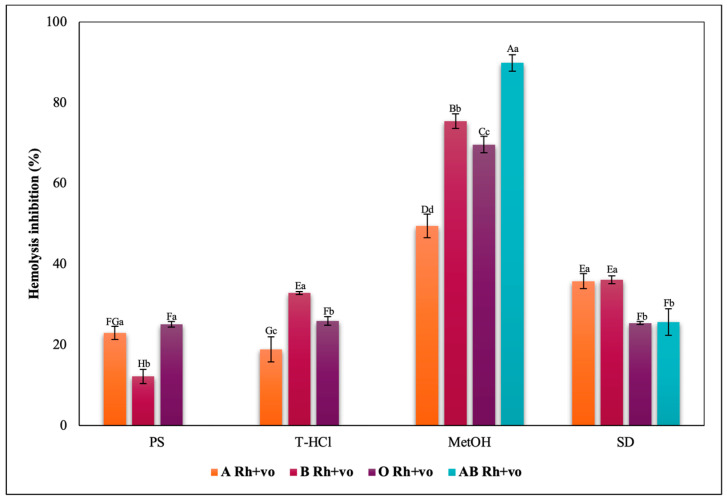
Inhibitory effect of heat-induced hemolysis of *P. cruentum* extracts on human erythrocytes with different ABO blood groups. Different letters represent significant differences in the means between treatments and groups (*p* ≤ 0.001). Capital letters represent a two-way ANOVA. Lowercase letters represent a univariate ANOVA between the extracts. PS, extract of physiological solution; T-HCl, 0.5 M Tris-HCl extract; MetOH, 99% methanol extract; SD, sodium diclofenac.

**Figure 18 metabolites-13-00366-f018:**
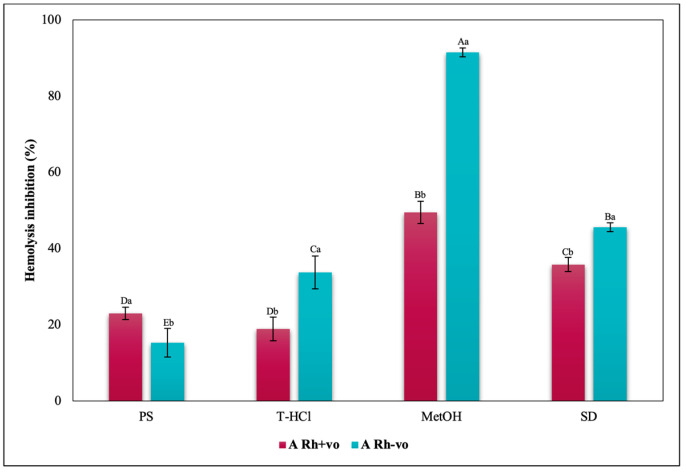
Inhibitory effect of heat-induced hemolysis of *P. cruentum* extracts on human erythrocytes with different RhD (positive and negative) of A blood group. Different letters represent significant differences in the means between treatments and groups (*p* ≤ 0.001). Capital letters represent a two-way ANOVA. Lowercase letters represent a univariate ANOVA between the extracts. PS, extract of physiological solution; T-HCl, 0.5 M Tris-HCl extract; MetOH, 99% methanol extract; SD, sodium diclofenac.

**Figure 19 metabolites-13-00366-f019:**
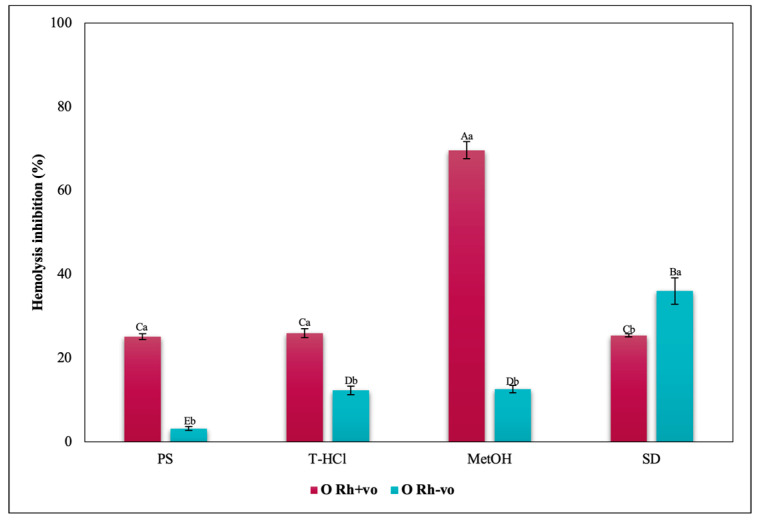
Inhibitory effect of heat-induced hemolysis of *P. cruentum* extracts on human erythrocytes with different RhD (positive and negative) of O blood group. Different letters represent significant differences in the means between treatments and groups (*p* ≤ 0.001). Capital letters represent a two-way ANOVA. Lowercase letters represent a univariate ANOVA between the extracts. PS, extract of physiological solution; T-HCl, 0.5 M Tris-HCl extract; MetOH, 99% methanol extract; SD, sodium diclofenac.

**Figure 20 metabolites-13-00366-f020:**
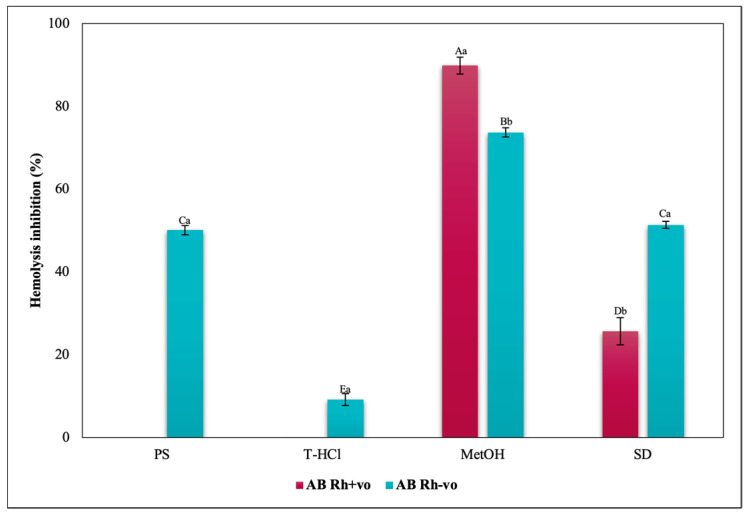
Inhibitory effect of heat-induced hemolysis of *P. cruentum* extracts on human erythrocytes with different RhD (positive and negative) of AB blood group. Different letters represent significant differences in the means between treatments and groups (*p* ≤ 0.001). Capital letters represent a two-way ANOVA. Lowercase letters represent a univariate ANOVA between the extracts. PS, extract of physiological solution; T-HCl, 0.5 M Tris-HCl extract; MetOH, 99% methanol extract; SD, sodium diclofenac.

**Table 3 metabolites-13-00366-t003:** Phycobiliprotein concentration of aqueous extracts from *P. cruentum* (mg·g^−1^ PS).

PBP	T-HCl	PS
B-PE	71.00 ^Aa^ ± 2.65	53.64 ^Ba^ ± 2.01
R-PC	37.56 ^Ab^ ± 0.87	30.60 ^Bb^ ± 1.90
APC	32.21 ^Ac^ ± 1.14	30.14 ^Ab^ ± 2.87

Data are shown as the mean ± SD (standard deviation) of at least three replicates (n ≥ 3). One-way analysis of variance (ANOVA). Different capital letters represent significant differences in the concentrations of phycobiliproteins in the extracts (*p* < 0.001). Different lowercase letters represent significant differences in the phycobiliprotein concentrations of each extract (*p* < 0.001). PS, extract of physiological solution; T-HCl, 0.5 M Tris-HCl extract; B-PE, B-Phycoerythrin; R-PC, R-Phycocyanin; APC, allophycocyanin.

**Table 4 metabolites-13-00366-t004:** Total chlorophylls and carotenoids in the methanolic extract of *P. cruentum* (mg·g^−1^ PS).

Pigment	MetOH
C_total_	7.78 ^a^ ± 0.63
C_x+c_	0.49 ^b^ ± 0.14

Data are shown as the mean ± SD (standard deviation) of at least three replicates (n ≥ 3). One-way analysis of variance (ANOVA). Different letters represent significant differences in the concentrations of pigments in the extracts (*p* < 0.001). MetOH, 99% methanol extract; C_total,_ Total chlorophylls; C_x+c,_ Total carotenoids.

**Table 5 metabolites-13-00366-t005:** Protein concentration of the aqueous extracts from *P. cruentum* (μg·mL^−1^).

Extract	Protein Concentration
PS	45.65 ^a^ ± 0.46
T-HCl	16.66 ^b^ ± 0.75

Data are shown as the mean ± SD (standard deviation) of at least three replicates (n ≥ 3). One-way analysis of variance (ANOVA). Different letters represent significant differences in the concentrations of pigments in the extracts (*p* < 0.001).

**Table 6 metabolites-13-00366-t006:** Antioxidant activity from *P. cruentum* extracts (µmol TE·g^−1^ PS).

Extract	ABTS^+•^	DPPH^•^	FRAP
PS	1238.49 ^Aa^ ± 20.51	344.22 ^Bc^ ± 05.82	335.57 ^Bc^ ± 06.68
T-HCl	1186.64 ^Bb^ ± 18.24	1385.65 ^Aa^ ± 22.04	374.30 ^Cb^ ± 27.30
MetOH	761.80 ^Bc^ ± 21.00	1318.72 ^Ab^ ± 14.52	433.10 ^Ca^ ± 43.40

Data are shown as the mean ± SD (standard deviation) of at least three replicates (n ≥ 3). One-way analysis of variance (ANOVA). Different capital letters represent significant differences in the concentrations of phycobiliproteins in the extracts (*p* < 0.001). Different lowercase letters represent significant differences in the phycobiliprotein concentrations of each extract (*p* < 0.001).

## Data Availability

The original contributions data presented in this research are included in the article; further inquiries can be directed to the corresponding authors.

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
