# Peer review of "Erythroprotective Potential of Phycobiliproteins Extracted from Porphyridium cruentum"

_metabolites, 2023, doi:10.3390/metabo13030366_

Round 1
Reviewer 1 Report
Comments and Suggestions of reviewer
The present manuscript «Erythroprotective potential of phycobiliproteins extracted from 2 Porphyridium cruentum » treats with interest the erythroprotective potential of phycobiliproteins from P. cruentum.
We can observe the seriousness with which the authors treated the experiment. However, a number of mentions should be made:
Comment
· In the abstract, authors should define all the abbreviations such as “ HAT and SET (line 24).
· Line 53 « Oceans are a promising source of natural resources, due to their extension, 53 little exploration, and biological richness [12], including microalgae », Please add examples of other microalgae cited by recent references known by these bioactive molecules.
· Line 68, after [17] remove « . »
· In this part « 2.2. Microalgal culture and experimental design » it is important to add the chemical composition of your microalgal biomass after lyophilization.
· Line 120 « 2.4. Phycobiliproteins extraction »and line 130 « 2.5. Phycobiliproteins extraction and pigments quantification » authors must check these 2 paragraphs, there is a repetition.
· Line 128 « Protein concentration was estimated quantitatively by the bicinchoninic acid assay (BCA) 128 [20] ». Authors should add the protocol with more details.
· Line 187 « obtained. In a 96-well microplate. The » Please check this sentence.
· In Fig 1 line 296 authors should add the error bar for all the points and correct the unit in the title.
· In table 1, please define all the abbreviations.
· Line 344, please check the spaces.
· Figure 6, lines 518: the standard deviation between the values is very high. Authors should repeat the experience of the blood biocompatibility assay of P. cruentum extracts on human erythrocytes.
· In the conclusion « P. cruentum » instead of « P. cruentum »
Author Response
Comment
- In the abstract, authors should define all the abbreviations such as “ HAT and SET (line 24).
Response: The comment was considered. See line 24-25.
- Line 53 « Oceans are a promising source of natural resources, due to their extension, 53 little exploration, and biological richness [12], including microalgae », Please add examples of other microalgae cited by recent references known by these bioactive molecules.
Response: The comment was considered. See line 77-81.
- Line 68, after [17] remove « . »
Response: The comment was considered. The word "after" was removed.
- In this part « 2.2. Microalgal culture and experimental design » it is important to add the chemical composition of your microalgal biomass after lyophilization.
Response: The authors appreciate the reviewer's comment. However, for the purposes of this research, it was considered to directly quantify the proteins of the extracts using the BCA method, since the objective of this research was focused on evaluating the erythroprotective potential of phycobiliproteins. See line 178
- Line 120 « 2.4. Phycobiliproteins extraction »and line 130 « 2.5. Phycobiliproteins extraction and pigments quantification » authors must check these 2 paragraphs, there is a repetition.
Response: The comment was considered.
- Line 128 « Protein concentration was estimated quantitatively by the bicinchoninic acid assay (BCA) 128 [20] ». Authors should add the protocol with more details
Response: The comment was considered. Added the requested information. See line 178-187.
- Line 187 « obtained. In a 96-well microplate. The » Please check this sentence.
Response: The comment was considered. The sentence was corrected. See line 209.
- In Fig 1 line 296 authors should add the error bar for all the points and correct the unit in the title.
Response: The comment was considered. See line 327.
- In table 1, please define all the abbreviations.
Response: The comment was considered. See line 341-343.
- Line 344, please check the spaces.
Response: The comment was considered. See line 378
- Figure 6, lines 518: the standard deviation between the values is very high. Authors should repeat the experience of the blood biocompatibility assay of P. cruentum extracts on human erythrocytes.
Response: The comment was considered. See line 574.
- In the conclusion « P. cruentum » instead of « P. cruentum »
Response: The comment was considered.

Reviewer 2 Report
It is a robust work on the erythroprotective effect of extracts and bioactives of an alga, however some points deserve emphasis and correction:
- Key terms and acronyms for understanding the text were not explained, nor was it described in full in its first presentation, such as the term HAT. Describes it and explained initially.
- The author reports in the course of the discussion paragraphs that methanol is a limiting factor in performance and a probable bias in the results, so knowing this, why did they not perform lyophilization as even suggested by the authors and thus avoid this methodological bias? (l. 562-568)
- The author despite adopting a great p of significance for his data, in several graphs a huge error (SD) can be seen as seen in several bars on the graphs, I suggest redoing the statistic and removing such data that deviate from the normality pattern , and with that to have reliable results after the statistical analysis. (Fig. 6, 13, 14 are highlighted).
- Considering that the represented antigens do not correspond to the total of erythrocyte antigens, it would be important to carry out the evaluation with red blood cells from the determination of other antigen groups, such as Duff, MN, Kell, LW,for example.
Author Response
Comment
- Key terms and acronyms for understanding the text were not explained, nor was it described in full in its first presentation, such as the term HAT. Describes it and explained initially.
Response: The comment was considered. See line 24-25.
- The author reports in the course of the discussion paragraphs that methanol is a limiting factor in performance and a probable bias in the results, so knowing this, why did they not perform lyophilization as even suggested by the authors and thus avoid this methodological bias? (l. 562-568)
Response: The authors are grateful for the reviewer's comments. The authors suggest rotoevaporation of methanol to avoid cytotoxicity, however this methodology will be considered for future research.
- The author despite adopting a great p of significance for his data, in several graphs a huge error (SD) can be seen as seen in several bars on the graphs, I suggest redoing the statistic and removing such data that deviate from the normality pattern , and with that to have reliable results after the statistical analysis. (Fig. 6, 13, 14 are highlighted).
Response: The comment was considered. See line 590, 908 and 953.
- Considering that the represented antigens do not correspond to the total of erythrocyte antigens, it would be important to carry out the evaluation with red blood cells from the determination of other antigen groups, such as Duff, MN, Kell, LW, for example.
Response: The authors appreciate the reviewer's comment. The ABO and RhD system antigens are the most immunogenic of all erythrocytes. These antigens present a great importance in transfunctional medicine. In addition, this ABO and RhD system are associated with different non-communicable chronic diseases (See line 53-56). Therefore, the Duff, MN, Kell, LW systems, among others, were not considered for study because they did not have this high immune response. However, your suggestion will be considered for future studies.

Reviewer 3 Report
In this manuscript, phycobiliproteins extracted from Porphyridium cruentum were characterized and evaluated their effects on erythrocytes. Finally, the authors claimed that phycobiliproteins have potential in hemolytic inhibition.
1. The writing of manuscript is complicated and a concise version is better for reading and understanding.
2. Line 13. “There are multiple associations between the different blood groups (ABO and RhD) and the incidence of oxidative stress related diseases.”. Please specify the sentence what types of oxidative stress related diseases.
3. Line 14. “Its prevention and treatment have been oriented towards the study of bioactive agents capable of combating its effects.” The sentence is not readable. What is the rationale?
4. Line 15. “Phycobiliproteins (PBP) are promising bioactive compounds present in the microalga Porphyridium cruentum, whose inhibitory effect on hemolysis has not been reported.”. Please specify why the effects of phycobiliproteins on hemolysis should be studied.
5. Line 24. “The results of the AAPH, hipotonicity and heat induced hemolysis revealed a probable relationship between the different antigens (ABO and RhD) with the antihemolytic effect, clarifying the biodirected drugs importance to treat and or prevent diseases.”. Please specify what types of diseases being treated or prevented.
6. The introduction of phycobiliproteins and Porphyridium cruentum, along with their biological implications is of importance. Besides, based on what rationale, biological studies of phycobiliproteins were emphasized on hemolysis.
7. 2.6. “their effects on the in vitro induction of the respiratory burst” was mentioned. However, there was any mention in the results.
8. Line 897. “The evaluation of the P. cruentum extracts erythroprotective potential revealed a probable relationship between the different antigens (ABO and RhD) with the antihemolytic effect. The importance of these phenomena elucidates the creation of drugs biotargeted to a certain blood group, to treat and/or prevent communicable and non-communicable diseases efficiently, in favor of human health.”. Please specify how to translate current study towards such conclusion.
Author Response
Comment
- “There are multiple associations between the different blood groups (ABO and RhD) and the incidence of oxidative stress related diseases.”. Please specify the sentence what types of oxidative stress related diseases.
Response: The comment was considered. See line 13-15.
- Line 14. “Its prevention and treatment have been oriented towards the study of bioactive agents capable of combating its effects.” The sentence is not readable. What is the rationale?
Response: The comment was considered. See line 14-15.
- Line 15. “Phycobiliproteins (PBP) are promising bioactive compounds present in the microalga Porphyridium cruentum, whose inhibitory effect on hemolysis has not been reported.”. Please specify why the effects of phycobiliproteins on hemolysis should be studied.
Response: The comment was considered. See line 15-17.
- Line 24. “The results of the AAPH, hypotonicity and heat induced hemolysis revealed a probable relationship between the different antigens (ABO and RhD) with the antihemolytic effect, clarifying the bidirected drugs importance to treat and or prevent diseases.”. Please specify what types of diseases being treated or prevented.
Response: The comment was considered. See line 25-27.
- The introduction of phycobiliproteins and Porphyridium cruentum, along with their biological implications is of importance. Besides, based on what rationale, biological studies of phycobiliproteins were emphasized on hemolysis.
Response: The comment was considered. Tables 1 and 2 were added to emphasize what was stated. See line 25-27.
- “their effects on the in vitro induction of the respiratory burst” was mentioned. However, there was any mention in the results.
Response: This statement was removed as it was not part of the research. It was a mistake, we apologize.
- Line 897. “The evaluation of the P. cruentum extracts erythroprotective potential revealed a probable relationship between the different antigens (ABO and RhD) with the antihemolytic effect. The importance of these phenomena elucidates the creation of drugs biotargeted to a certain blood group, to treat and/or prevent communicable and non-communicable diseases efficiently, in favor of human health.”. Please specify how to translate current study towards such conclusion.
Response: The comment was considered. See line 1173-1177

Round 2
Reviewer 1 Report
We can observe the seriousness with which the authors treated the experiment.
I think that the manuscript has been improved but a number of mentions should be made to warrant publication in “Metabolites”:
· In table 3, you should correct" clorophylls" by chlorophylls
· In table 4 you should correct "aqueus extract" by "aqueous extract extract"
· It is necessary to add the location of the figures in the text
· I think that the result in figure 6 is not convincing
· Authors have to check the forme and the language used.
Author Response
- In table 3, you should correct" clorophylls" by chlorophylls
Response: The comment was considered. See line 345.
- In table 4 you should correct "aqueus extract" by "aqueous extract".
Response: The comment was considered. See line 401.
- It is necessary to add the location of the figures in the text
Response: The comment was considered. See line 532-533, 657, 666, 1049, 1060, and 1154.
- I think that the result in figure 6 is not convincing
Response: The authors appreciate the reviewer's comment. The Blood biocompatibility assay on human erythrocytes aims to evaluate the safety of various types of phytochemicals. Among them, the phycobiliproteins of P. cruentum. The blood biocompatibility of the methanolic and aqueous extracts on group A erythrocytes was analyzed during a 6-h exposure period. The assay revealed that only the methanolic extract presented erythrocyte hemolysis (2 to 2.6 %, approximately) in exposures of 4 and 6 hours, while the aqueous extracts did not present hemolysis (0 %). The test was repeated several times obtaining the same results. Therefore, it is suggested that the blood biocompatibility of bioactive compounds depends on the erythrocyte surface antigen. In previous publications it has been observed that blood biocompatibility and the bioactivity of phytochemical compounds depend on erythrocyte surface antigens, as well as an association between ABO and RhD blood groups with erythroprotective potential (https://doi.org/10.3390/metabo12121203).
- Authors have to check the form and the language used.
Response: The comment was considered. A comprehensive revision of English was carried out.

Reviewer 2 Report
Due to the adjustments made by the authors, I consider the deficiencies remedied and the study is suitable for publication.
Author Response
Due to the adjustments made by the authors, I consider the deficiencies remedied and the study is suitable for publication.
Response.
The authors appreciate their time spent on the review.
